# Increased Risk of Hospitalization for Lower Respiratory Tract Illness (LRTI) in the Elderly Living in Communities near a Newly Established Industrial Estate in Central Thailand

**DOI:** 10.3390/ijerph22060874

**Published:** 2025-05-31

**Authors:** Pirawan Wangupadcha, Steven Ronsmans, Jeroen Vanoirbeek, Yuparat Limmongkon, Wipharat Phokee, Chananya Jirapornkul

**Affiliations:** 1Faculty of Public Health, Khon Kaen University, Khon Kaen City 40002, Thailand; pirawan.occ@gmail.com; 2Center for Environment and Health, Department of Public Health and Primary Care, KU Leuven, 3000 Leuven, Belgium; steven.ronsmans@kuleuven.be (S.R.); jeroen.vanoirbeek@kuleuven.be (J.V.); 3Department of Occupational Safety and Environmental Health, Khon Kaen University, Khon Kaen City 40002, Thailand; yupali@kku.ac.th (Y.L.); pwipha@kku.ac.th (W.P.); 4Department of Epidemiology and Biostatistics, Khon Kaen University, Khon Kaen City 40002, Thailand

**Keywords:** hospitalization, lower respiratory tract illness, industrial emissions, industrial estate, chronic disease

## Abstract

Air pollution and health effects in communities near industrial areas are matters of concern in Thailand. Elevated air pollution concentrations are associated with morbidity and mortality arising from lower respiratory tract illness. In this study, we aimed to assess whether living in a community near a new industrial estate in Central Thailand was associated with an elevated incidence of lower respiratory tract illness (LRTI) among the elderly. We used data on hospital admissions from a primary hospital and sub-district-level health-promoting hospitals in six sub-districts located within a radius of 5 km from the industrial estate, between 1 November 2023 and 31 March 2024, using a cross-sectional study design. There were 1311 elderly individuals living within a radius of 3 km and 1488 between 3 and 5 km. We found a statistically significant increased risk of hospitalization for LRTI in the elderly living within a radius of less than 3 km (odds ratio: 3.43 [95% CI: 1.61–7.36] compared to those living in a radius between 3 and 5 km. Our results provide strong support for an association between hospitalization for LRTI and living distance from a new industrial estate.

## 1. Introduction

Exposure to air pollution is a global issue that directly affects respiratory health. In the Global Burden of Disease Study, it was estimated that in 2019, exposure to air pollution was responsible for 695,000 deaths and 15.4 million disability-adjusted life years (DALYs) due to chronic obstructive pulmonary disease (COPD) and 326,000 deaths and 19.5 million DALYs due to lower respiratory tract infections worldwide [1]. In Asia, between 2010 and 2017, approximately 21.4 million people died from chronic respiratory diseases, although with large differences in mortality among countries. In Thailand, the age-standardized crude mortality rate from chronic respiratory diseases during this period was around 20 per 100,000 inhabitants per year [2].

Air pollution exposure has a direct impact on the increase in lower respiratory tract illness, especially in communities near industrial activity [1]. Lower respiratory tract disease is strongly associated with air pollution. An increase of 10 μg/m^3^ in the two-year average of PM_2.5_ is associated with a 2.24% increase in the incidence of asthma [3]. NO_2_ is associated with daily admissions for lung infections, asthma, and COPD, and PM_2.5_ is associated with COPD hospital admissions [1]. In China, each 10 µg/m^3^ increase in PM_2.5_ is associated with a 3.51% (95% CI: 0.96–6.12%) increase in the risk of admission for acute lower respiratory infection admission. It has been shown that admissions for lower respiratory tract diseases increase as PM_2.5_ concentrations increase [4].

Air pollution has been found within a radius of 3 km from industrial areas, and excess risk (OR: 95% CI) has been found near (≤3 km) the production of metals (2.66: 1.77–4.00), surface treatment of metals (1.48: 1.08–2.02), and production of glass and mineral fibers (2.06: 1.39–3.07). Also, an excess risk (OR: 95% CI) of colorectal cancer has been detected near industries overall for all distances analyzed, from 1 km (2.03: 1.44–2.87) to 3 km (1.26: 1.00–1.59) [5]. People living within 3 km of an industrial area exhibited the largest effects via respiratory symptoms compared to those at further distances; evidence from coal-fired brick kilns in Bangladesh indicated 2.2 (95% CI: 1.2–4.3) greater odds of COPD symptoms among adults over 40 and 4.2 (95% CI: 2.7–6.8) greater odds of respiratory symptoms among adults over 18 [6]. People living near the Industrial Estate Islamabad in the industrial sector of the Islamabad Capital Territory in Pakistan (within a radius of approximately 650 m) had an increased risk of lower respiratory tract disease, including chronic bronchitis, phlegm, and dyspnea [7]. Residents living near a Petrochemical Industrial Complex in seven countries—Argentina (La Plata), Brazil (Rio Grande do Norte), Taiwan (Miaoli, Jenwu, Linyuan, Chunghua, and Yunlin), Thailand (Rayong Province), Spain (Tarragona), Italy (Sardinia and Basilicata), and the United Kingdom (Teesside)—had a significantly higher incidence of cough, wheezing, bronchitis, rhinitis, and asthma [8]. The prevalence rates of asthma and rhinoconjunctivitis were found to be 24% and 34%, respectively, among people living near gold mines and copper mines in Northern Chile, at average distances of 2.1 and 1.9 km, respectively [9]. Higher levels of PM_2.5_ and NO_X_ from industry emissions have also been associated with decreased lung function, including a lower forced vital capacity (FVC) and a lower forced expiration volume in 1 s (FEV_1_), and have been associated with dry cough [10]. The ambient particulate matter pollution-related burdens of LRTI and COPD were higher among countries with low and low–middle socio-demographic indices (SDIs), while countries with high–middle SDIs showed the highest burden of tracheal, bronchus, and lung cancer attributable to exposure to ambient particulate matter pollution [1]. People living in developing countries, including Southeast Asia, are increasingly burdened with outdoor air pollution. This is partly an adverse result of the drive of the industrial sector to develop the economy of the country [11].

Thailand is one of those countries in Southeast Asia in which the industrial sector is developing rapidly, becoming the heart of the economy. An industrial estate (or park) is an area that is shared by many producers who partly share the same infrastructure and facilities. Industrial estates have become important contributors to the industrial development of the country [12]. Currently, there are 66 such locations throughout all regions of Thailand with various types of industries. They form an important industrial production base in the Association of Southeast Asian Nations (ASEAN), but, at the same time, the air quality and community health are negatively affected, especially in factory areas. NO_2_ concentrations in the industrial area of Rayong have shown to be 10.4–34.0 µg/m^3^, significantly higher than in urban and background areas [13]. The average PM_2.5_ concentrations are 38.8 ± 25.2 µg/m^3^ in the wet season and 52.7 ± 33.9 µg/m^3^ in the dry season [14]. In Nakhon Ratchasrima, the mean value of the 24 hr average PM_2.5_ concentration in the industrial area was 34.7 µg/m^3^, significantly higher than in urban areas [15]. Communities surrounding Map Ta Phut Industrial Estate in Rayong Province face air pollution problems and have a higher incidence of respiratory diseases due to air pollution [14,16]. Residents living near industrial areas suffer from respiratory diseases due to air pollution. Children living near industrial areas in Rayong, Thailand, particularly those areas with high levels of pollutants, are at risk of chronic bronchitis, bronchial asthma, dyspnea and wheezing, persistent cough, and persistent phlegm. There is a higher prevalence in communities near industrial areas than in distant communities. Children in schools within 1 km of a petrochemical industry area have been shown to have a substantially higher risk of experiencing respiratory symptoms (OR 3.41 [1.70–6.85]) than those in schools between 5 and 10 km from this industry [17]. Air pollution concentrations in the central and surrounding areas in the past 10 years (2013–2022) have begun to improve, but there are still some pollutants that exceed the standard, especially PM_2.5_ and NO_2._, and NO_2._ from November to March in 2022 showed the highest 1 h average at each measurement point in the range of 21–189 ppb (average of 63 ppb). The annual average was in the range of 2–30 ppb, average 10 ppb, and was found to exceed the standard in two areas in the central region, with the highest number of industrial factories [18]. The highest PM_2.5_ concentration over the 24 h average ranged between 22 and 156 µg/m^3^. In 2023, the PM_2.5_ level in Bangkok and its vicinity become more severe, representing a worsened image for the overall area. The PM_2.5_ level reached an average of 31 μg/m^3^, and there were 97 days on which it exceeded the standard [19].

The global population aged 60 years and older is expected to double in the next 35 years. Caring for this elderly population has become an important global healthcare issue. In 2016, more than 1 million people aged 70 years and older died from infections of the lower respiratory tract, with increasing age being the largest factor that increases morbidity and death rates. Advanced age is associated with reduced physiological reserves and an aging immune system [20]. Currently, Thailand is a country with an aging society; in 2023, there were 13.64 million people aged over 60 years (19.5%) [21]. However, there have been no studies on air pollution and its effects on the health of the elderly living in areas near relatively new industries in Thailand. Therefore, in the present study, we aimed to study the effect of a new industrial estate in central Thailand, established less than ten years ago, on air pollution and the health of the elderly population living nearby. We measured the concentrations of PM_2.5_ and NO_2_ and evaluated the association between the hospitalization of elderly people for LRTI and living in communities near this new industrial estate. The results of this study will lead to the determination of measures to prevent and control air pollution and its impact on the health of surrounding communities so that the same problems do not repeat themselves near industrial estates that have been established for longer periods of time. It will also make industry leaders with factory expansion plans aware of the need for adequate air pollution control measures.

## 2. Materials and Methods

### 2.1. Hospitalization Data

We included 2799 elderly subject from the six sub-districts examined in this study. We collected the following data on these elderly patients from their medical files: reason for hospitalization (lower respiratory tract illness or not), gender, age, body mass index, education level, current occupation, smoking status, history of drinking alcohol, and chronic diseases (diabetes, hypertension, dyslipidemia, etc.).

To evaluate the association between, on the one hand, the incidence of hospital admissions due to LRTI among the elderly and, on the other hand, living near the industrial estate, LRTI data were obtained from 43 health database files of each of six sub-district health-promoting hospitals and one district hospital in the period from 1 November 2023 to 31 March 2024. Using the International Classification of Diseases, Revision 10 (ICD-10), all healthcare contacts due to acute or chronic lower respiratory symptoms by individuals residing in the selected sub-districts were identified. ICD-10 J20-22 codes were used to identify acute LRTI and ICD-10 J40-47 codes to identify chronic LRTI.

Because not all covariables were available for all sub-districts (smoking data, especially, were lacking for three sub-districts), we divided the data analysis into two parts: in part one, we analyzed the entire elderly population in the study area (2799 elderly people from the six sub-districts), and in part two, we analyzed only the data of the sub-districts for which all variables were available. The selected sub-districts for part two were representative of the nearby as well as the remote area population (1347 elderly people from three sub-districts).

### 2.2. Study Area: Communities near a New Industrial Estate

The study area includes 6 sub-districts in the central region of Thailand. We selected the sub-districts according to distance from a new industrial estate, i.e., areas located within a radius of 5 km around the estate. Communities were divided into those located “near” (0–3 km) and those located “far” (>3–5 km) from the fence the new industrial estate. The people living 3–5 km away from the industrial site were the control group for those living < 3km away. The control group was chosen as the 2 groups are very similar regarding socioeconomic and other factors (except for nearness to the industrial site).

### 2.3. Measurement of Air Pollutants

We measured PM_2.5_ and NO_2_ concentrations in the communities within 3 km and those between 3 and 5 km from the new industrial estate. We measured in 4 areas from 17 to 25 January 2024, continuously for 24 h a day, covering working days and weekends. Measurements were performed according to the US EPA standard [22,23], using a low-volume PM_2.5_ air sampler (model: Partisol 2000i, 2000i203361302, Thermo Scientific (Waltham, MA, USA) and TE-Wilbur, 0133, Tisch Environmental (Cleves, OH, USA), flow rate: 16.67 (L/min), air pressure: 759.5 (mmHg)) and an NO_x_ Chemiluminescence Analyzer (model: Thermo Electron (Waltham, MA, USA), 42C-0508011076, air pressure: 759.5 (mmHg)).

### 2.4. Statistical Data Analysis

We investigated the associations between hospitalization for LRTI and distance from a new industrial estate using a cross-sectional analytical study design. We used univariable logistic regression analysis to explore unadjusted associations between hospitalization for LRTI and various factors. The data are presented with crude odds ratios, 95% confidence intervals of the crude odds ratios, and *p*-values. Results with *p*-values < 0.25 were selected for the initial multivariable model.

Multivariable logistic regression analysis was used to investigate the associations between hospitalization for LRTI and distance from the new industrial estate, adjusted for various factors. The analysis was performed separately for the entire population and for the subgroup for which all variables were available. The suitability of the model was evaluated using quality of fit. We set the significance level at 0.05. The final model is presented with adjusted odds ratios, 95% confidence intervals of the adjusted odds ratios, and *p*-values.

### 2.5. Ethics Statement

This study has been reviewed by the Khon Kaen University Ethics Committee for Human Research based on the Declaration of Helsinki and The ICH Good Clinical Practice Guidelines; the date of approval is 7 November 2023.

## 3. Results

### 3.1. Study Area and Measured Air Pollutants

Figure 1 shows the six sub-districts surrounding the new industrial estate. We found that among the six sub-districts, five had an average PM_2.5_ concentration exceeding the WHO air quality guidelines and one had a concentration below guidelines (Table 1).

### 3.2. Descriptive Statistics of the Study Population

A total of 2799 elderly people (54.38% female) lived in the study area, with more than half aged between 60 and 69 years (59.16%), 68.66% with normal BMI, 46.84% living within 3 km of the new industrial estate, and 75.46% living downwind of it (when considering north-east to be the prevailing wind direction for this time of the year) (Table 2). The incidence rate of hospitalization for LRTI over the 5-month study period was 2.36%. For the sub-districts for which the complete dataset was available, we found that 77.43% of the elderly had at least one chronic disease, while 6.98% had high blood pressure and 6.09% had diabetes. Most of them were currently not working (anymore) (95.55%), and most had completed only primary school education (95.32%). Only 10.10% were current or former cigarette smokers.

### 3.3. The Association Between the Incidence of LRTI Among Elderly People and Living in Communities near the New Industrial Estate

In the univariable analysis, we found that the elderly living in communities near the industrial estate (0–3 km) had higher rates of hospitalization for LRTI (OR 2.15, 95% CI: 1.15–4.06) than those living further away (3–5 km) from the estate (Table 3). In the multivariable analysis, we adjusted for various potential confounders: gender, age, BMI, education, smoking, and diabetes. The multivariable analysis confirmed the higher rates of hospitalization for LRTI in the elderly living near the industrial estate (OR 3.43; 95% CI: 1.61–7.36, *p*-value: 0.001).

Other factors were also associated with increased rates of hospitalization in the multivariable model. Being underweight (BMI < 18.50 kg/m^2^; OR 5.39; 95% CI: 2.00–14.50, *p*-value: 0.001) or overweight (BMI ≥ 23.00 kg/m^2^; OR 1.89; 95% CI: 0.85–4.15, *p*-value: 0.112) led to higher rates of hospitalization. In addition, there was an increased rate of hospitalization for LRTI among the elderly with diabetes (OR 33.89; 95% CI: 15.01–76.05, *p*-value: <0.001). Compared with those between 60 and 69 years old, subjects aged 70–79 (OR 1.33; 95% CI: 0.57–3.07, *p*-value: 0.508) and those over 80 years of age (OR 2.50; 95% CI: 1.02–6.10, *p*-value: 0.045) had higher rates of hospitalization for LRTI.

## 4. Discussion

Our study is the first to examine air pollution concentrations in communities near an industrial estate that is no more than 10 years old, and to assess air pollution and its health impact on the elderly. Elderly people are a vulnerable group in this new residential environment, in which former agricultural communities have transformed into communities near industrial estates. We expect that there will be more communities like this due to economic expansion in many countries around the world.

Previous studies have suggested that communities within a radius of 5 km from industrial zones should be monitoring for air pollution and its health effects. We measured the air concentration of PM_2.5_ in the area closest to a new industrial estate (0–3 km) and an area further away. Both areas were similar in that the PM_2.5_ concentration was higher than the WHO air quality guidelines (<37.5 µg/m^3^). There was no significant difference in PM_2.5_ concentration between the two areas. During the time we measured PM_2.5_, the wind speed was very low (5.99 Knots), or no air movement was observed. It may be that there is no difference in the PM_2.5_ concentration in both areas. Weather conditions could have affected the measured concentrations of PM_2.5_ [24,25,26].

We found a statistically significant increase in the risk of hospitalization for LRTI in the elderly living within a radius of 3 km (OR 3.43 (95% CI: 1.61–7.36)). This is consistent with several previous studies that found an increased risk of respiratory diseases associated with residential proximity to industrial zones. Adults living in communities less than 650 m away from the Industrial Estate Islamabad in Pakistan had a significant increase in chronic bronchitis, phlegm, wheezing, and dyspnea [7]. In Northern Chile, communities near gold mines and copper mines had a high prevalence of asthma [9]. A study in seven countries—Argentina (La Plata), Brazil (Rio Grande do Norte), Taiwan (Miaoli, Jenwu, Linyuan, Chunghua, and Yunlin), Thailand (Rayong Province), Spain (Tarragona), Italy (Sardinia and Basilicata), and the United Kingdom (Teesside)—found that the incidence of bronchitis, asthma, cough, wheezing, and rhinitis were associated with living near a petrochemical industrial complex [8]. Exposure to industrial air pollution in residencies less than 5 km from industrial areas has been associated with respiratory symptoms, phlegm, cough [11,26], and lower lung function parameters (FEV_1_, FVC) [10]. Among adults living within a 5 km radius of an industrial area in Malaysia, 34.5% experienced coughing and 25.5% chest tightness [11]. In New York State, a study found strong associations between living within 5 km of a biorefinery facility and respiratory emergency department visit rates related to (exacerbations of) emphysema, asthma, chronic bronchitis, and chronic airway obstruction [27].

We found significant evidence of an increased risk of hospitalization for LRTI among elderly people with diabetes (OR 33.98; 95% CI: 15.01–76.05). This is consistent with a study in Japan that found significant associations between diabetes and mortality from respiratory infection [14]. Also, significant associations have been found between diabetes and mortality from respiratory infection among both men and women (HR 1.39; 95% CI, 1.10–1.76 and HR 2.30; 95% CI, 1.71–3.11, respectively) [24]. It was also found that being either underweight (<18.50 kg/m^2^) or overweight (≥23.00 kg/m^2^) was associated with hospitalization for LRTI in the elderly. For the underweight elderly, we found an OR of 5.39 (95% CI: 2.00–14.50), and for the overweight an OR of 1.89 (95% CI: 0.85–4.15) compared with the normal-weight elderly. Also, increasing age is an important factor that increases morbidity and mortality. It is associated with reduced physiological reserves and an aging immune system. Worldwide, in 2016, more than 1 million people aged 70 years or over died from lower respiratory tract infections [20]. In the USA, it was found that increasing age is associated with the incidence of emphysema to the extent of 0.11 and 0.06 for every 2 μg/m^3^ unit increase in PM_2.5_ and 10 ppb NO_x_ [25]. Increasing age is associated with respiratory disease, characterized by increased inflammation and decreased lung function, including asthma and COPD [28]. The results from our study support age as an important risk factor for hospitalization due to LRTI. For those older than 80 years, we found an OR of 2.50 (95% CI: 1.02–6.10) compared to those aged 60 to 69 years.

We found an association between having had higher education and having been hospitalized for LRTI (OR 3.06 [95% CI: 0.95–9.91]). We assume that this association results from the fact that education leads to better job opportunities and higher income, making it easier to access healthcare services [29,30]. Smoking is an important risk factor for severe LRTI; our study confirmed that cigarette smoking increased the risk of hospitalization for LRTI in the elderly (OR 1.41 [95% CI: 0.40–5.01]). Previous studies have found that current smokers were twice as likely to have asthma and that there is an increased risk of respiratory disease in smokers who have chronic disease [17,27,31,32,33].

A major strength of our study is that we were able to describe both air pollution and health effects. Firstly, we measured PM_2.5_ and NO_2_ in the community (Table 1). Secondly, we analyzed the association between distance between communities and the new industrial estate. The results of our study strongly support the proposition that nearby areas are more at risk. Nevertheless, because PM_2.5_ and NO_2_ concentrations were similar in all measured areas, we were unable to identify the causal exposures that led to the elevated risk of hospitalization due to LRTI in the elderly living near the industrial estate. It is likely that industrial emissions which we did not measure could explain the associations that we found in this study. Although the findings reflect conditions in a specific industrial region in central Thailand, they may offer relevant insights for other rapidly developing industrial areas with similar environmental and demographic contexts. This study contributes important evidence supporting the integration of environmental health considerations into industrial development policies. To protect at-risk populations, particularly the elderly, we recommend incorporating health risk assessments into industrial estate planning and design. This may include the implementation of health-protective zoning, buffer zones, green barriers, and enhanced ventilation designs for nearby housing. Additionally, routine health screenings, improved access to healthcare services, and community-based air quality monitoring programs can help mitigate health risks and support vulnerable communities.

## 5. Conclusions

We found that the concentration of PM_2.5_ in areas within 0–3 km of the industrial area was not different from that of the area beyond it. However, it was found that elderly people living near industrial areas were more affected in terms of health. It was found that elderly people who live near industrial areas have a higher incidence of hospitalization due to LRTI than elderly people who live just a little further away. In addition, having diabetes mellitus, a body mass index less than normal, and a higher education level is also associated with risk of hospitalization due to LRTI.

Elderly people who live near industrial areas are a high-risk group that requires screening and health monitoring, especially for lower respiratory system diseases. Especially during the dusty season, from November to March, it is crucial for local health service units to monitor dust concentration levels in the air. This is important for vulnerable populations, such as elderly individuals with diabetes or those with an abnormal body mass index (BMI). Given their increased susceptibility to respiratory issues, these groups should also undergo regular screening for lower respiratory tract diseases, which can be exacerbated by air pollution.

A limitation of our study is the incompleteness of the secondary data from 43 health databases on many variables, such as individual exposure levels, which can vary due to daily activity patterns, time spent indoors, housing characteristics, socioeconomic status (SES), occupational history, length of living in the community, family history of chronic disease, day of admission and date of discharge from the hospital, and the precise symptoms of lower respiratory tract illness. In addition, we studied only high-risk communities in areas that are within a 5 km radius of industrial estates. There was no comparison group in this remote area. This is also a limitation of this study. A future study should collect primary data on the indoor air samples at their houses, individual exposure levels, and primary data on the health effects on the elderly, such as respiratory function data (FEV1/FEVC) for initial symptoms of lower respiratory system disorders: coughing, wheezing, gasping, shortness of breath, etc. The Health Risk Increase (HRI) from air pollutants data and Health Burden from HRI should also be calculated. This will provide information about symptoms and important factors that affect lower respiratory tract illnesses.

## Figures and Tables

**Figure 1 ijerph-22-00874-f001:**
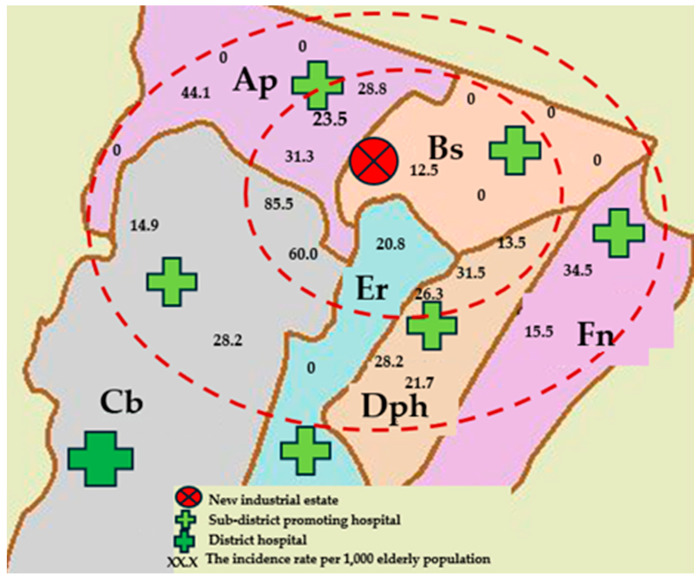
Map showing the new industrial estate in central Thailand, the six surrounding sub-districts, the location of the hospitals, and the incidence rate of hospitalization for LRTI in the elderly population.

**Table 1 ijerph-22-00874-t001:** Descriptive statistics of PM_2.5_ and NO_2_ concentrations (18–25 January 2024).

PM_2.5_					NO_2_			
Distance	Min	Median	Mean (SD)	Max	Min	Median	Mean (SD)	Max
Near (0–3 km)								
Ap	23.1	34.3	32.5 (6.8)	41.4	0.004	0.005	0.005 (0.0002)	0.005
Bs	17.5	41.6	38.8 (10.1)	49.4	0.012	0.013	0.013 (0.0003)	0.013
Far (>3–5 km)								
Cb	29.9	34.6	38.3 (7.6)	50.9	0.004	0.004	0.004 (0.0005)	0.005
Dph	32.6	41.3	44.6 (10.7)	58.5	0.002	0.002	0.002 (0.0004)	0.003

SD: Standard deviation.

**Table 2 ijerph-22-00874-t002:** Baseline characteristics of the study population.

Factors	Entire Region (n = 2799)		Subregion (n = 1347)	
	Number	Percentage (%)	Number	Percentage (%)
LRTI				
No	2733	97.64	1305	96.88
Yes	66	2.36	42	3.12
Gender				
Female	1522	54.38	745	55.31
Male	1277	45.62	602	44.69
Age (years)				
60–69	1656	59.16	776	57.61
70–79	721	25.76	344	25.54
≥80	422	15.08	227	16.85
Mean (±SD)	69.48 (±8.36)		69.71 (±8.43)	
Median (Min:Max)	67 (60:113)		68 (60:97)	
BMI (kg/m^2^)				
Normal weight (18.50–22.99 kg/m^2^)	1187	68.66	660	49.00
Underweight (<18.50 kg/m^2^)	152	10.66	124	9.20
Overweight (≥23.00 kg/m^2^)	460	20.66	563	41.80
Mean (±SD)	22.84 (±2.84)		22.73 (±3.61)	
Median (Min:Max)	23.11 (10.59:41.87)		22.35 (10.59:41.87)	
Chronic disease				
N/A	1428	51.02	0	0
Any chronic disease	1043	37.26	1043	77.43
Hypertension	104	3.72	94	6.98
Diabetes	89	3.18	82	6.09
Education level				
N/A	1428	51.02	0	0
Primary school	1305	46.62	1284	95.32
Higher than primary school	66	2.07	63	4.68
Occupation				
N/A	1428	51.02	0	0
None	1287	46.38	1287	95.55
Farmer	36	1.29	36	2.67
Laborer	26	0.93	12	0.89
Other	4	0.14	10	0.74
Smoking cigarettes				
N/A	1428	51.02	0	0
No	1235	44.12	1211	89.90
Yes (current or previous)	136	4.86	136	10.10
Drinking alcohol				
N/A	1428	51.02	0	0
No	1235	44.12	1066	79.14
Yes (current or previous)	136	4.86	281	20.86
PM_2.5_ average concentration (µg/m^3^)				
Low (<37.5 µg/m^3^)	698	24.95	0	0
High (≥37.5 µg/m^3^)	2101	75.05	1347	100
NO_2_ average concentration (ppm)				
Low (<0.17 ppm)	2799	100	1347	100
High (≥0.17 ppm)	0	0	0	0
Wind direction				
Upwind	687	24.54	285	21.16
Downwind	2112	75.46	1062	78.84
Distance from home to the new estate				
Far (>3–5 km)	1488	53.16	760	56.42
Near (0–3 km)	1311	46.84	587	43.58

**Table 3 ijerph-22-00874-t003:** Univariable and multivariable analysis of the associations between hospitalization for lower respiratory tract illness (LRTI) and distance from a new industrial estate, adjusted for various factors, for a subregion (n = 1347).

Factors	Number	LRTI		Univariable Analysis			Multivariable Analysis		
	n	n	(%)	OR	95% CI	*p*-Value	OR_adj_	95% CI	*p*-Value
Overall	1347	42	3.12						
Distance						0.017			0.001
Far (>3–5 km)	760	16	2.11	1			1		
Near (0–3 km)	587	26	4.43	2.15	1.15–4.06		3.43	1.61–7.36	
Gender						0.100			0.173
Female	745	18	2.42	1			1		
Male	602	24	3.99	1.67	0.90–3.12		1.80	0.89–3.67	
Age (Years)						0.296			0.021
60–69	776	21	2.71	1			1		
70–79	344	10	2.91	1.08	0.50–2.31	0.850	1.33	0.57–3.07	
≥80	227	11	4.85	1.83	0.87–3.86	0.112	2.50	1.02–6.10	
BMI						0.001			0.280
Normal weight	660	12	1.82	1			1		
Underweight	124	11	8.87	5.25	2.26–12.20	<0.001	5.39	2.00–14.50	
Overweight	563	19	3.37	1.88	0.90–3.92	0.089	1.89	0.85–4.15	
Education						0.055			0.001
Primary	1284	37	2.88	1			1		
Higher than primary	63	5	7.94	2.90	1.10–7.66		3.06	0.95–9.91	
Smoking cigarettes						0.899			0.648
No	1211	38	3.14	1			1		
Yes (current or former)	136	4	2.94	0.94	0.33–2.66		1.41	0.40–5.01	
Diabetes						<0.001			<0.001
No	1265	22	1.74	1			1		
Yes	82	20	24.39	18.22	9.45–5.16		33.89	15.01–76.05	

## Data Availability

No new data were created in this study. Data sharing is not applicable to this article.

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
