# Peer review of "Increased Risk of Hospitalization for Lower Respiratory Tract Illness (LRTI) in the Elderly Living in Communities near a Newly Established Industrial Estate in Central Thailand"

_ijerph, 2025, doi:10.3390/ijerph22060874_

Round 1
Reviewer 1 Report (Previous Reviewer 1)
Comments and Suggestions for Authors
Hi everyone,
In my opinion, the changes you made really helped make your article stronger and more solid.
Well done!
Author Response
Thank you for your comments.
Reviewer 2 Report (Previous Reviewer 2)
Comments and Suggestions for Authors
Discussion of the results is insufficient, rest is ok.
Author Response
Comment: The English could be improved to more clearly express the research.
Response:
Thank you for your suggestion regarding the English language. The manuscript has been carefully revised by all co-authors, who have extensive experience in writing and publishing in English-language journals. We believe the current version meets the expected linguistic standards. Nonetheless, we have re-read the manuscript attentively to ensure clarity and correctness throughout.
I have revised the manuscript by adding a discussion and suggestions for future research, in accordance with your recommendations. The details are provided in the revised version attached herewith.

This manuscript is a resubmission of an earlier submission. The following is a list of the peer review reports and author responses from that submission.
Round 1
Reviewer 1 Report
Comments and Suggestions for Authors
Hello,
Congratulations on the study conducted.
1. The objective of the study is both current and relevant, with some degree of innovation.
The abstract immediately informs the reader about the study's purpose and conclusions, which is highly important.
2. The introduction provides sufficient, relevant, and necessary information for a better understanding and evaluation of the results and discussion presented by the authors.
3. Regarding the methodology (section 2), in my opinion, the sequence of presentation would be clearer if section 2.3 were introduced first:
“We included 2,799 elderly individuals from the six sub-districts in this study. We collected the following data on these elderly individuals from their medical records: reason for hospitalization (lower respiratory tract illness or not), gender, age, body mass index, education level, current occupation, smoking status, history of alcohol consumption, and chronic diseases (diabetes, hypertension, dyslipidemia, etc.).”
After this, the first paragraph of this section should follow:
“To evaluate the association between...,”
and then sections 2.1 and 2.2.
For section 2.2, I believe it would be helpful to include more detailed information about the equipment used to measure PM2.5 and NO2 concentrations.
4. As for the results, they are clearly presented.
5. In the discussion, it would be valuable if the authors tried to justify why the concentration of PM2.5 in areas close to the industrial zone (0-3 kilometers) was not significantly different from the adjacent areas.
6. In the conclusion, you state:
“A future study should, in addition to collecting primary data on air samples, also collect primary data on the health effects of the elderly.”
In your opinion, what type of data would be relevant to include in future studies? For example, respiratory function data (FEV1) or other parameters? If so, which ones?
Great work!
Author Response
Comments 1: Regarding the methodology (section 2), in my opinion, the sequence of presentation would be clearer if section 2.3 were introduced first:
“We included 2,799 elderly individuals from the six sub-districts in this study. We collected the following data on these elderly individuals from their medical records: reason for hospitalization (lower respiratory tract illness or not), gender, age, body mass index, education level, current occupation, smoking status, history of alcohol consumption, and chronic diseases (diabetes, hypertension, dyslipidemia, etc.).”
After this, the first paragraph of this section should follow:
“To evaluate the association between...,”
and then sections 2.1 and 2.2.
For section 2.2, I believe it would be helpful to include more detailed information about the equipment used to measure PM2.5 and NO2 concentrations.
Response 1, Thank you for pointing this out. We agree with this comment. Therefore, we have changed the sequence of presentation which the section 2.3 were introduction first then 2.1 and 2.2 (page no.3 , paragraph 2, and line123-155).
For section 2.2, We have specified the flow rate, air pressure and model of the PM2.5 and NO2 measuring device. (page no.4 , paragraph 2, and line156-159).
Comments 2; In the discussion, it would be valuable if the authors tried to justify why the concentration of PM2.5 in areas close to the industrial zone (0-3 kilometers) was not significantly different from the adjacent areas.
Response 2, Thank you for pointing this out. We agree with this comment. Therefore, we have added wind speed data while measuring PM2.5 concentration, which is a factor that affects PM2.5 concentration (page no.8 , paragraph 2, and line 257-159) .
Comments 3; In the conclusion, you state:
“A future study should, in addition to collecting primary data on air samples, also collect primary data on the health effects of the elderly.”
In your opinion, what type of data would be relevant to include in future studies? For example, respiratory function data (FEV1) or other parameters? If so, which ones?
Response 3, Thank you for pointing this out. We agree with this comment. Therefore, We have added details that should be studied further regarding the measurement of PM2.5 and parameters that indicate lower respiratory tract symptoms. (page no.10 , paragraph2, and line335-339) .
Reviewer 2 Report
Comments and Suggestions for Authors
I thoroughly read the manuscript entitled “Increased Risk of Hospitalization for Lower Respiratory Tract Illness (LRTI) of Elderly Living in Communities near a Newly Established Industrial Estate in Central Thailand.” Although the title of the manuscript is very interesting, it does not contain significant results. Only table 1 contains some results for particulate matters and NO2; tables 2 and 3 contain demographic data. The manuscript does not contain risk analysis as mentioned in the title. The author should provide significant results for the publication of the manuscript. The structure of the manuscript is good and well written.
Comments on the Quality of English Language
Remove the grammatical mistakes.
Author Response
Comment :I thoroughly read the manuscript entitled “Increased Risk of Hospitalization for Lower Respiratory Tract Illness (LRTI) of Elderly Living in Communities near a Newly Established Industrial Estate in Central Thailand.” Although the title of the manuscript is very interesting, it does not contain significant results. Only table 1 contains some results for particulate matters and NO2; tables 2 and 3 contain demographic data. The manuscript does not contain risk analysis as mentioned in the title. The author should provide significant results for the publication of the manuscript. The structure of the manuscript is good and well written.
Response, Thank you for pointing this out. This study examines “The increased risk of hospitalization for lower respiratory tract illness (LRTI) of elderly living in communities near a newly established industrial estate in central Thailand”. We found a statistically significant increased risk of hospitalization for LRTI in the elderly living within a radius of less than 3 kilometers (odds ratio 3.43 [95%CI: 1.61- 7.36]), compared to those living in a radius between 3 and 5 kilometers. Which in table 3, page number 7, section 3.3, the multivariable analysis confirmed the higher rates of hospitalization for LRTI in the elderly living near the industrial estate (OR 3.43; 95%CI: 1.61-7.36, p-value: 0.001). Therefore, our results provide strong support for the associations between hospitalization for LRTI and living distance from the new industrial estate.
Round 2
Reviewer 2 Report
Comments and Suggestions for Authors
Authors fails to answer the question; results could be significant if author provide the control site data. WITHOUT control site results correlation between industrialization and hospitalization is not considered significant.
Heatlh risk analysis are still missing, author must calculate Helth Risk Increase (HRI) from air pollutants data. Also calculate Health Burdan from HRI.
Comments on the Quality of English Language
Ok
Author Response
Comments 1: Authors fails to answer the question; results could be significant if author provide the control site data. WITHOUT control site results correlation between industrialization and hospitalization is not considered significant.
Response 1, Thank you for pointing this out. We agree with this comment. Therefore, We have added reference studies of air pollution emissions within 1-3 kilometers from industry and the largest public health impacts within 3 kilometers as cut-off points. (page no.2 , paragraph 2, and line 52-63) .
Comments 2: Health risk analysis are still missing, author must calculate Helth Risk Increase (HRI) from air pollutants data. Also calculate Health Burdan from HRI.
Response 2, Thank you for pointing this out. We agree with this comment. We have added health risk calculations as a recommendation of necessary future studies. (page no.10 , paragraph 2, and line 338-339) .
Comments 3: A control population is needed to compare with the exposed population
Response 3, Thank you for pointing this out. We agree with this comment. We have added the issue of comparison groups as one of the limitations of our study. (page no.10 , paragraph 2, and line 332-335).